# The Effect of Cooling Rate from Solution Treatment on γ′ Reprecipitates and Creep Behaviors of a Ni-Based Superalloy Single-Crystal Casting

**Jiapeng Huang** [1], **Cheng Ai** [2], **Yi Ru** [1], **Yong Shang** [1], **Yanling Pei** [1], **Shusuo Li** [1], **Shengkai Gong** [1] **and Heng Zhang** [1,*]

1 School of Material Science and Engineering, Beihang University, Beijing 100191, China
2 School of Materials Science and Engineering, Chang'an University, Xi'an 710064, China
* Correspondence: hengzz@buaa.edu.cn

**Abstract:** Slowing down the coarsening of the γ′ phase and suppressing the precipitation of the topologically close-packed (TCP) phase is crucial for optimizing the creep properties of Ni-based single crystal superalloys, which are affected by the solution treatment history. In the present study, the effect of cooling rate on the morphology, size and lattice misfit of γ′ reprecipitates after solution treatment, as well as the microstructural evolution (e.g., the coarsening of γ′ reprecipitate and precipitation of the TCP phase) and creep properties of samples under different cooling rates at 1100 °C were investigated. The findings suggested that as the cooling rate increasing, the size of γ′ reprecipitates decreased, while the morphology transformed from polygonal to cubic. Additionally, faster cooling rate, on the one hand, the lower the degree of lattice misfit of γ/γ′ phases, which is beneficial to slow down the coarsening of the γ′ phase; on the other hand, the supersaturation of the γ-phase was reduced, i.e., the Mo and Re contents in the γ matrix were lower compared to the slow-cooling sample, which led to a lower TCP phase area fraction during thermal exposure at 1100 °C. It is crucial that the creep life of the alloy significantly improved by increasing the cooling rate after solution treatment; this is facilitated by the formation of rafts from finer cubic γ′ phase and reduction in the TCP phase as a source of microcracks. In conclusion, the findings of this study provide new insights into suppressing the precipitation of the TCP phase and optimizing alloy heat treatment processes to improve creep properties.

**Keywords:** Ni-based superalloy; single-crystal casting; γ′ reprecipitates; solution treatment; cooling rate; TCP; creep



## 1. Introduction

Ni-based single crystal (SX) superalloys are extensively used for modern areo-engines due to their excellent microstructural stability, mechanical properties and environmental tolerance at elevated temperatures [1,2]. Generally, the excellent mechanical properties of Ni-based SX superalloys at elevated temperature derives from their unique two-phase equilibrium microstructure, consisting of ordered $Ni_3Al$ (γ′ phase)-strengthening precipitates with an $L_{12}$ structure, which are coherently embedded in the face-centered cubic (FCC) matrix γ-phase [3,4]. In addition, high temperature creep strength is a key parameter with which to evaluate the high mechanical properties of Ni-based SX superalloys, which depends on the interaction of dislocations with the γ′ precipitates and interface of γ/γ′ phases [5–8]. However, the γ–γ′ microstructure undergoes degradation during the prolonged high-temperature exposure, including coarsening of γ′ precipitates and precipitation of the topologically close-packed (TCP) phase [9–11], which makes the superalloy enter the rapid deformation stage and fracture. Although a proper initial γ–γ′ microstructure is obtained, it is necessary to consider slowing down the coarsening of γ′

precipitates and inhibiting the precipitation of TCP phase, which are very important for improving the high-temperature creep properties of the alloy.

The coarsening of $\gamma'$ precipitates is driven by the reduction in the interfacial energy and coherent strain energy of the $\gamma/\gamma'$ phases [12], where the former is related to its size and the latter is related to the distribution of alloying elements in the $\gamma/\gamma'$ phases. In addition, whether the TCP phase can be precipitated depends on the enrichment degree of Re, Mo, Cr and other elements in the $\gamma$ matrix of the superalloy [13,14]. Therefore, reducing the degree of segregation of alloying elements in dendrites structure or $\gamma/\gamma'$ phases is beneficial for slowing down the coarsening of $\gamma'$ precipitates and inhibiting the precipitation of the TCP phase.

Generally, a proper heat treatment is required before service for Ni-based SX superalloys to eliminate the $\gamma$–$\gamma'$ eutectic, reduce the microsegregation of elements such as Re, Mo, Cr [15] and optimize the size and morphology of the $\gamma'$ reprecipitations (secondary $\gamma'$ phase) [16], thereby obtaining excellent properties for the alloy [17–19]. Among them, an appropriate solution treatment (ST) process (including austenitizing temperature, holding time and cooling rate, etc.) is the basis for subsequent microstructural evolution and regulation. Elimination of residual microsegregation is usually performed by increasing the austenitizing temperature or prolonging the holding time, but this will lead to the formation of more and larger homogenized pores [20,21], so that the purpose of improving the high-temperature creep performance cannot be achieved; furthermore, Hegde [22] and Karunaratne et al. [23] suggested that the microsegregation does not change monotonically with the increase in the solution temperature and the prolongation of the solution time, and the microsegregation of Re, Cr and Mo elements increased due to the uphill diffusion of the $\gamma$–$\gamma'$ eutectic in the inter-dendrites. However, it has been reported that by increasing the cooling rate after solution treatment from supersolvus temperature, on the one hand, the size of the $\gamma'$ reprecipitates becomes smaller [24] (the interface energy of $\gamma$–$\gamma'$ phases is larger), but its morphology changes from cubic to irregular shape. Similar results were reported by Bhowal [7] and Yu [25]; on the one hand, they report how the secondary $\gamma'$ precipitates with non-equilibrium compositions formed at high cooling rate due to the limited time available for diffusion and reduction in mobility of elements [26,27]. Indeed, this is beneficial to reduce the degree of elemental segregation between $\gamma$- and $\gamma'$ phases, as confirmed by the work of Conner [28] and Mitchell [29]. Nonetheless, the high-temperature coarsening behavior of $\gamma'$ reprecipitates with non-equilibrium components is still unclear at present. Secondly, the effects of $\gamma$–$\gamma'$ microstructure evolution and the precipitation behavior of the TCP phase at higher cooling rates after solution treatment on high-temperature creep properties are currently lacking systematic studies.

The purpose of this paper is to investigate the effect of cooling rates (water quenching and air cooling) after solution treatment from supersolvus temperature on the coarsening of $\gamma'$ reprecipitates, the precipitation behavior of the TCP phase and high-temperature creep properties (1100 °C/137 MPa). The variation in size and morphology of the $\gamma$–$\gamma'$ phase microstructure and microsegregation behavior under different cooling rates were analyzed in detail, and the relationship between the microstructure evolution and the creep behavior was discussed. The findings of this study provide new insights into suppressing precipitation of TCP and optimizing alloy heat treatment processes to improve creep properties.

## 2. Materials and Methods

### 2.1. Materials and Heat Treatment

A low-density experimental nickel-based single crystal superalloy designed with high $\gamma'$ volume fraction and Mo content was adopted in this work. It had relatively good high-temperature mechanical properties, although there is a certain tendency of TCP precipitation. The chemical composition (weight percent) of the experimental superalloy (Exp. Alloy) is listed in Table 1, and was determined and verified by inductively coupled plasma–mass spectrometry (ICP-MS).

**Table 1.** Chemical composition of the experimental superalloy, wt.%.

| Element | Al | Mo | Re | Ta | Hf | Y | Ni |
|---------|-----|------|------|--------|------|-------|------|
| nominal | 7.0~8.5 | 7.0~11.0 | 3.0 | 1.0~4.5 | 0.1 | 0.05 | Bal. |
| measured | 6.9~8.3 | 7.2~11.3 | 2.85 | 1.2~4.6 | 0.08 | 0.045 | Bal. |

The single-crystal cylindrical rods (about φ16 × 150 mm) were prepared by a high-rate solidification (HRS) Bridgman apparatus using the screw-selecting method with the withdraw rate of 75 μm/s, the temperature gradient was approximately 4 K/mm, and the rods were chosen according to the orientations only within 5° deviating from the [001] orientation determined with the Laue X-ray back reflection method. In addition, the dendrite structure had the same orientation, and the grain boundaries were not observed after macro-etching, which means that the cylindrical rods had a single crystal structure. In order to eliminate the primary phases and reduction in the segregation to produce a homogeneous microstructure, a solution heat treatment was applied to dissolve the precipitated phases for subsequent reprecipitates in an optimized morphology and size. A heat treatment (HT) regime included a multi-stage solution treatment (1310 °C/2 h + 1320 °C/4 h + 1330 °C/4 h + 1340 °C/6 h + 1350 °C/6 h) and two-step aging treatment (1040 °C/2 h + 870 °C/32 h) was carried out in a tubular resistance furnace. Different cooling rates after ST were obtained by two cooling methods. One is to put the sample directly into water (20 °C) after removing it from the furnace (labeled as WQ), which has a cooling rate of about 1000–1200 K/min; the other (labeled as AC) is to cool the sample in flowing air at a cooling rate of 150~300 K/min. The detailed heat treatment process is listed in Table 2. To observe the microstructure evolution during high-temperature thermal exposure, the samples were subjected to an isothermal exposure of 1100 °C and rapidly water quenched to maintain the situation at high temperatures.

**Table 2.** Heat treatment schedule of experimental alloy under different cooling rates.

| Sample | Heat Treatment Schedule | Cooling Rate after ST (K/min) |
|--------|------------------------|-------------------------------|
| AC | 1310 °C/2 h + 1320 °C/4 h + 1330 °C/4 h + 1340 °C/6 h + 1350 °C/6 h, air cooling | 150~300 |
| WQ | 1310 °C/2 h + 1320 °C/4 h + 1330 °C/4 h + 1340 °C/6 h + 1350 °C/6 h, water cooling | 1000~1200 |
| HT | 1310 °C/2 h + 1320 °C/4 h + 1330 °C/4 h + 1340 °C/6 h + 1350 °C/6 h, air cooling, 1040 °C/2 h, air cooling, 870 °C/32 h, air cooling | 150~300 |

Note: AC represents air cooling after solution treatment, WQ represents water cooling after solution treatment, HT represents solution treatment + aging treatment.

### 2.2. Creep Test

Creep tests were performed at 1100 °C/137 MPa on rods with orientation within 5° deviating from [001] and with the gauge length of 25 mm and diameter of 5 mm. The tests were interrupted after creep deformation for 10 h and cooled down to room temperature before removing the load to study the dislocation structure formed during creep. The strain was measured by a displacement sensor with the accuracy of $1 \times 10^{-3}$ mm. At least 3 specimens were tested at each condition.

### 2.3. Microstructural Characterization

The microstructures of the specimens were characterized by a scanning electron microscopy (SEM, ZEISS EVO10, Oberkochen, Germany). Measurements of the γ' phase area fraction were performed by image analysis on black and white images of samples that were quenched after heat treatment and creep. A JEM-2100 transmission electron microscope (TEM) with an auxiliary energy-dispersive spectroscopy (EDS) detector was used to observe the γ–γ' phase morphology and elemental distribution of the heat-treated samples and measure the dislocation structure after creep. The TEM samples were cut from cross-sections parallel to (001) of the experimental alloys and electrochemically thinned in a solution of 15 mL perchloric acid and 85 mL alcohol at −20 °C.

### 2.4. Lattice Misfit Determination

The lattice parameters of two phases at room temperature after different heat treatment are measured by using Rigaku SmartLab X-ray diffractometer (CuKα: 40 kV, 200 mA) with a scanning speed of 0.5°/min. The (200) peak of the nickel-based superalloy, which is a combination of the overlapping (200) peak of the γ matrix and the corresponding peak of the γ′ phase, was measured. Bragg's law states that when an X-ray beam impinges on a crystal lattice, the maximum intensity of the diffracted beam occurs when the angle between the incident and the diffracted beam $2\theta_{hkl}$ the wavelength of the radiation, $\lambda$, and the distance between lattice planes, $d_{hkl}$ are related by Equation (1):

$$\lambda = 2d_{hkl} \sin\theta_{hkl} \tag{1}$$

where, *hkl* denotes the Miller indices for the specific family of lattice planes, and in this study *hkl* = 200. Moreover, the lattice misfit $\delta$ was determined from the lattice parameters of the γ- and γ′ phases ($a_\gamma$ and $a_{\gamma'}$) according to Equation (2):

$$\delta = 2(a_{\gamma'} - a_\gamma)/(a_{\gamma'} + a_\gamma). \tag{2}$$

The measured X-ray profiles of the (200) reflections were then split up into three Gaussian peak profiles using Origin 9.0 Software.

### 3. Results

#### 3.1. Microstructure of the As-Cast and Heat-Treatmented Alloy

The as-cast microstructure is shown in Figure 1. It has a distinct "cross-flowered" dendritic structure arranged in the same direction, and no obvious grain boundaries are observed after macro-etching using optical microscope (OM), as shown in Figure 1a, which means that it was a single crystal structure. The interdendritic region was further examined with SEM, where the large bulky γ′ phase (primary γ′) and (Mo, Re)-rich phase were observed, as shown in Figure 1b, c. However, after ST treatment, the SEM backscattered electrons (BSE) morphology of alloy and elemental distribution map are shown in Figure 2. The bulky γ′ phase and (Mo, Re)-rich phase at interdendrites are completely dissolved, and the elements are basically uniformly distributed, except for the slight segregation of Re in the cores of the dendrites. Thus, it is believed that the adopted solid solution regimen is capable of achieving the goal of solid solution completely.

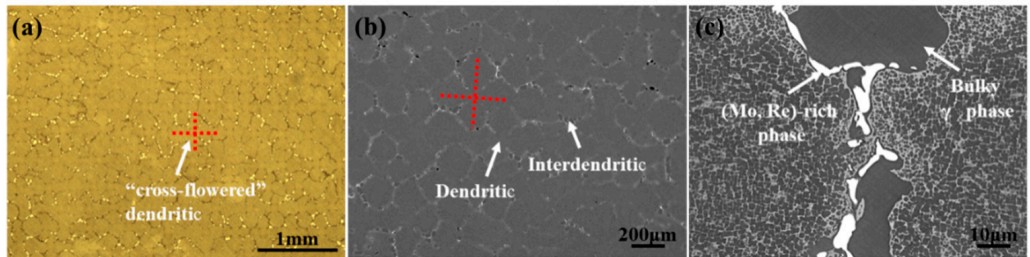

**Figure 1.** The microstructure of as-cast experimental alloy: (**a**) OM image of dendritic structure; (**b**) SEM of dendritic structure; (**c**) interdendritic region of (**b**).

The SEM secondary electrons (SE) morphology of γ′ reprecipitation after solution treatment under different cooling rates is displayed in Figure 3. The reprecipitated γ′ phase was uniformly distributed overall and embedded in the γ matrix channel. Nevertheless, the microstructure and size of the γ′ phase was significantly affected by the cooling conditions. With the decrease in cooling rate, the size of γ′ phase was bigger. After being cooled down by WQ and AC, the size of secondary γ′ precipitate was measured to have the statistical mean diameters of 0.359 and 0.491 μm, respectively, as shown in Figure 4a,b. Meanwhile, the γ-channel was transformed from a serrated to a straight one with the increasing the cooling rate. This came from the result of the growth of the γ′ phase and the increased

coherent strain energy. It is not unnoticeable that the lower the cooling rate is, the relatively larger the area fraction of $\gamma'$ phase, which increased from 75.6% for the WQ sample to 79.7% for the AC sample. The detailed statistics analyses are presented in Table 3. It can be assumed that the lower cooling rate provided for the precipitation and growth of the $\gamma'$ phase. The cooling process after solution treatment was equivalent to undergoing a transient aging treatment, and this is more clearly confirmed by comparing the results of the HT samples, as seen in Figure 3c. After two-step aging treatment, the size of the $\gamma'$ phase grew to 0.621 µm and its area fraction reached 82.3%, as listed in Table 3. However, the number density of the $\gamma'$ phase for the HT sample was about $6.4/\mu m^2$, which was much lower than that of $12.7/\mu m^2$ under the WQ cooling condition. Under slow-cooling conditions, the number density of $\gamma'$ phase is lower. This leads to a decrease in the total coherent strain energy and results in a coarsening of the $\gamma'$ phase [7]. Indeed, the secondary $\gamma'$ precipitates are transformed from sphere to cube with the slow-cooling rate [25].

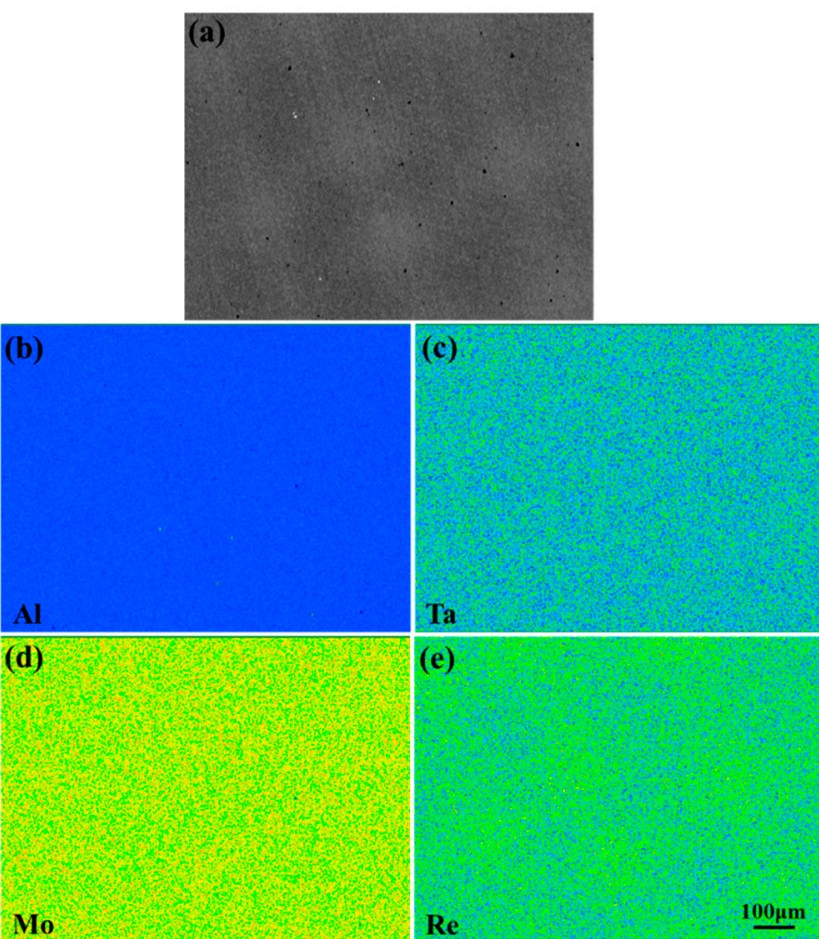

**Figure 2.** (**a**) The SEM-BSE morphology of alloy after ST treatment and elemental distribution map: (**b**) Al, (**c**) Ta, (**d**) Mo and (**e**) Re.

The microstructure of $\gamma'$ phases under different cooling conditions are shown in Figure 5. It is found that the finer $\gamma'$ phases are distributed orderly under WQ cooling condition, and the four sides are concave, as marked with a green dashed line in Figure 5a. This is due to the fact that the $\gamma'$ phase particles have the smallest elastic modulus along the [001] orientation; it will preferentially grow along the direction of body diagonals under the interaction of coherent strain. Meanwhile, the growth of cube corners will be enhanced due to the larger potential rate of supply of solute to these regions [30]. These $\gamma'$ phases are unstable, and as adjacent $\gamma'$ precipitates grow and collide with each other, the concave edges tend to be straightened due to the coherent strain energy [31]. In contrast, when the

cooling condition is AC, a large number of polygonal shape γ′ phase exist. Some vanishing γ-channels were located in the γ′ phase, which indicates that the growth of γ′ precipitates itself by merging adjacent to each other. In addition, the tertiary γ′ phases appeared in the γ-channel of the AC sample with a size of about 30~55 nm. After examining the composition of the tertiary γ′ phase, it is found that its Al, Ta content was lower compared with the secondary γ′ phase, but its Mo and Re contents were higher. It typically results that multiple nucleation events occurred at different undercooling degrees below the γ′ solvus due to a complex interaction between the increasing thermodynamic driving force of nucleation. This results from the temperature drop (increased undercooling) and the decreasing diffusivity of alloying elements and previous nucleation events resulting in a decrease in this driving force; this is similar to the results reported by Singh [27]. Surprisingly, for the HT sample, aside from the further coarsening of the γ′ precipitates, a dislocation network was found to be formed at the γ–γ′ interface. This is most likely due to a high amount of Mo and Re addition, causing a large lattice misfit. Even the spontaneous formation of dislocation networks to relaxing the interfacial misfit stress without an external stress.

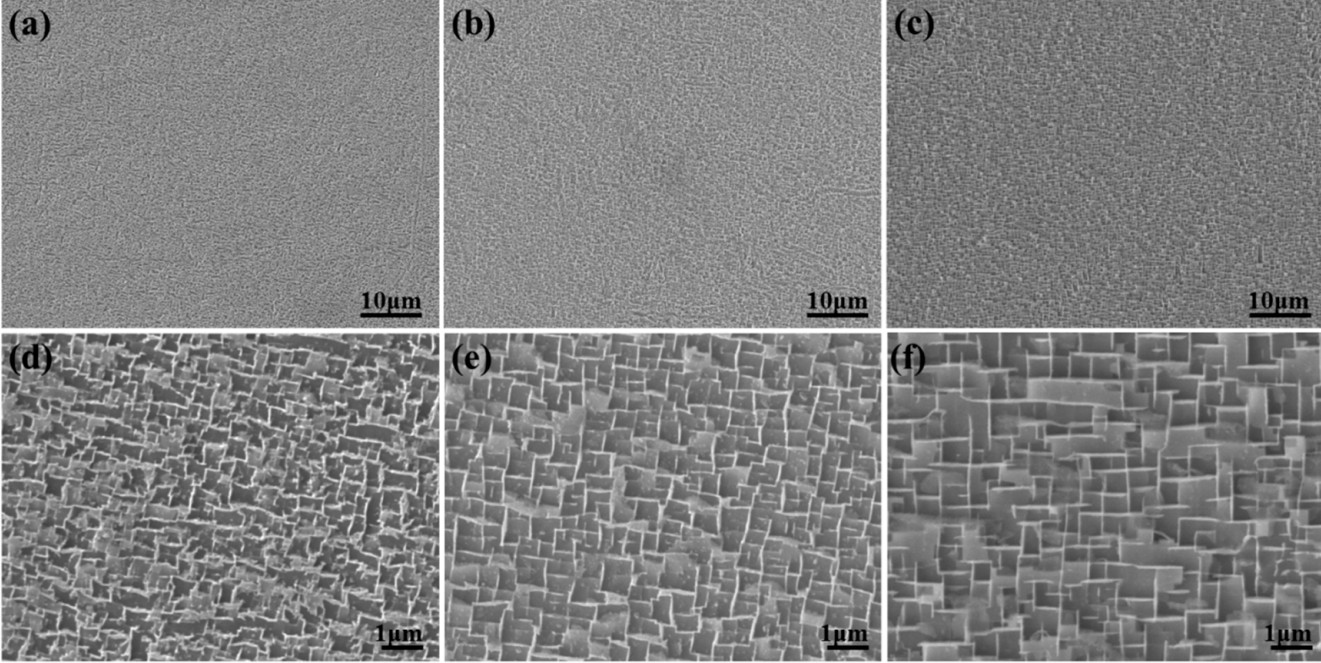

**Figure 3.** The SEM-SE morphology of secondary γ′ phases after heat treatment under different cooling rates: (**a**,**b**) WQ, (**d**,**e**) AC and (**c**,**f**) HT.

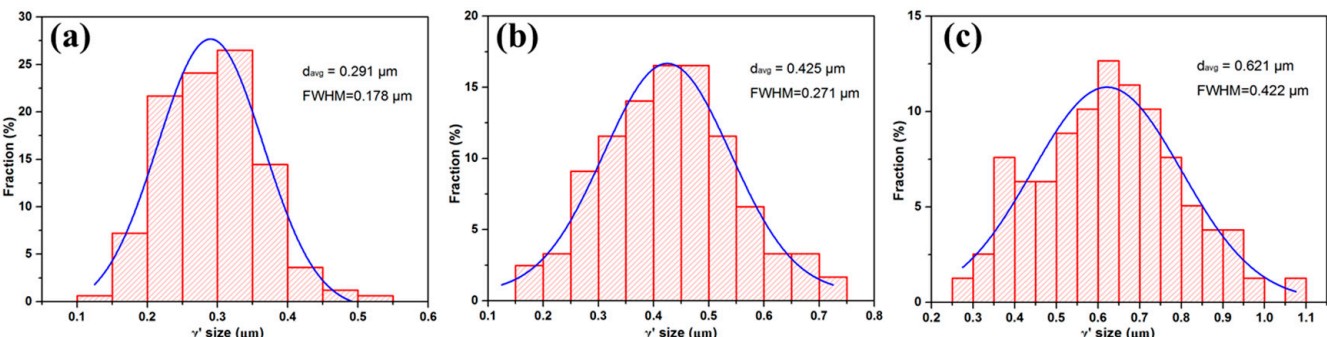

**Figure 4.** The size distribution of secondary γ′ precipitates after heat treatment under different cooling rates: (**a**) WQ, (**b**) AC and (**c**) HT.

**Table 3.** Size and area fraction of γ′ phase after heat treatment under different cooling rates.

| Sample | Size of γ′ Phase (μm) | Area Fraction of γ′ Phase (%) | Number Density of γ′ Phase (μm$^{-2}$) |
|---|---|---|---|
| HT | 0.621 | 82.3 ± 2.4 | 4.53 |
| AC | 0.425 | 79.2 ± 2.8 | 6.40 |
| WQ | 0.291 | 76.6 ± 3.2 | 12.78 |

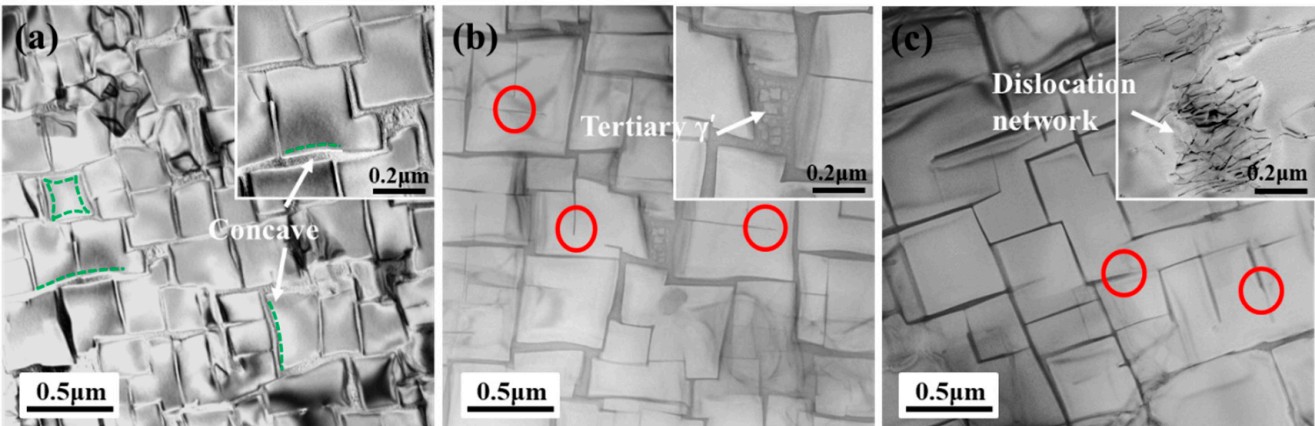

**Figure 5.** The TEM microstructure of γ′ phases after different heat treatment schedule: (**a**) WQ; (**b**) AC and (**c**) HT.

### 3.2. Element Partitioning and Lattice Misfit of γ–γ′ Phase

The compositions of γ- and γ′ phases under different cooling conditions and full heat treatment was measured by TEM equipped with EDX spectra. The results are listed in Table 4. It is evident that the reprecipitated γ′ precipitates with a lower Al and Ta contents while the γ-phase matrix with a lower Mo and Re contents in WQ condition compared to the AC condition. This was probably due to the fact of the formation of γ′ precipitates at higher temperatures during the slow-cooling condition, where the faster diffusion rates and longer diffusion time allowed these precipitates to achieve a near-equilibrium composition [32]. This result was enhanced after the two-step aging treatment in HT sample. For the quantitative assessment of the elemental distribution behavior, the elemental partitioning coefficient $k_i$ is introduced:

$$k_i = C_i^{\gamma'} / C_i^{\gamma} \tag{3}$$

where $C_i^{\gamma'}$ and $C_i^{\gamma}$ are the atomic fractions of an element i in the γ′ and γ-phases, respectively. From the calculation results of $k_i$ in Table 4, the elements of Mo and Re partition preferentially to the γ matrix with $k_i < 1$, while the elements Al and Ta partition preferentially to the γ′ precipitates with $k_i > 1$. It is clear that with the decrease in cooling rate, the partitioning of Mo and Re to the γ matrix and Al amd Ta to the γ′ precipitates is simultaneously enhanced. Therefore, those elements are soluble in γ matrix, Mo and Re should diffuse away, while elements Al and Ta should diffuse into the γ′ precipitate to accomplish the γ′ coarsening.

Figure 6 shows the fitted X-ray profiles of the (002)$_{\gamma, \gamma'}$ planes of alloy after ST under different cooling rate and HT at room temperature. It is clear that the peaks from γ- and γ′ phases of the AC sample are more spaced apart, and the diffraction angles shifted to lower values compared to the WQ sample, as seen in Figure 6a, b, which indicated that it has a larger absolute value of lattice misfit. However, the mismatch of the sample becomes smaller after aging, which is different from the Vegard Law calculation [33], as seen in Figure 6d, probably due to the coarsening of the γ′ phase and the release of the mismatch stress by the formation of the dislocation network.

**Table 4.** The compositions of $\gamma$- and $\gamma'$ phase under different cooling conditions and fully heat treatment (at.%).

| Condition | | Al | Mo | Re | Ta | Ni |
|---|---|---|---|---|---|---|
| WQ | $\gamma$ | 7.82 | 11.53 | 2.36 | 1.31 | 76.98 |
| | $\gamma'$ | 11.96 | 4.28 | 0.56 | 2.25 | 80.95 |
| | $k = C_{\gamma'}/C_{\gamma}$ | 1.53 | 0.37 | 0.24 | 1.71 | 1.05 |
| AC | $\gamma$ | 4.26 | 13.46 | 3.26 | 1.19 | 77.82 |
| | $\gamma'$ | 12.59 | 4.05 | 0.50 | 2.37 | 80.49 |
| | $k = C_{\gamma'}/C_{\gamma}$ | 2.96 | 0.30 | 0.15 | 1.98 | 1.03 |
| HT | $\gamma$ | 3.65 | 13.17 | 4.65 | 1.17 | 77.48 |
| | $\gamma'$ | 13.70 | 4.35 | 0.53 | 2.32 | 79.10 |
| | $k = C_{\gamma'}/C_{\gamma}$ | 3.75 | 0.33 | 0.11 | 1.99 | 1.02 |

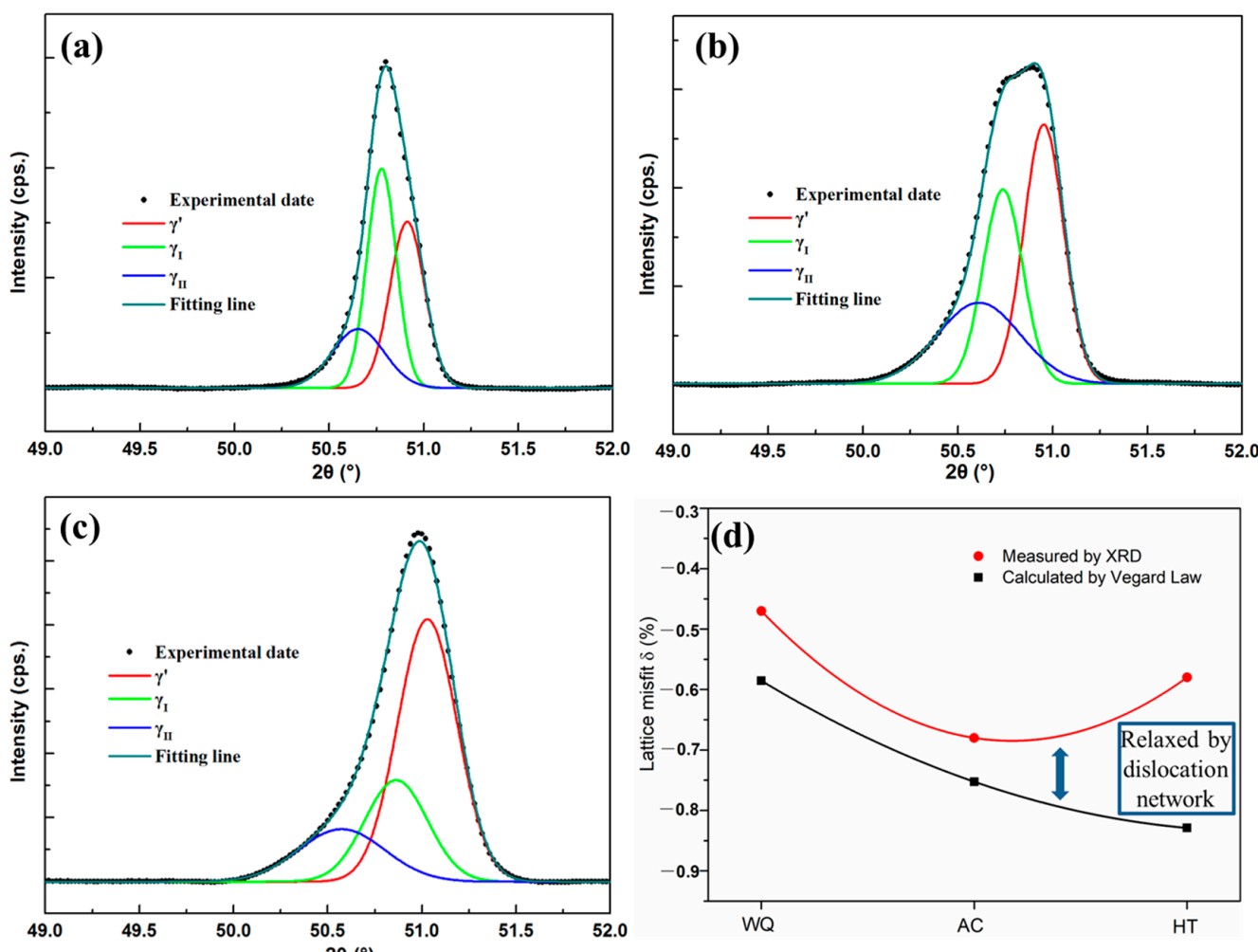

**Figure 6.** Diffractometer scans of (002) $\gamma$, $\gamma'$ peaks of experimental alloy after different heat treatment at room temperature: (**a**) WQ, (**b**) AC, (**c**) HT and (**d**) the lattice misfit measured by XRD and calculated by Vegard Law (the phase compositions in Table 4 were used).

### 3.3. Microstructure Evolution during Thermal Exposure at 1100 °C

Figure 7 illustrates the microstructure of $\gamma'$ and $\gamma$-phases in the alloy after different heat treatment during thermal exposure at 1100 °C for 1 h, 10 h and 50 h. With the prolonged thermal exposure time, the $\gamma'$ phase underwent undirected coarsening and the morphology transforms from cubic to irregular. It can be seen that the $\gamma'$ precipitates are uniformly distributed, and their morphology remains mostly cubic after 1 h of thermal exposure in

the WQ and AC samples. However, at the same time, the $\gamma'$ phase in the HT samples clearly underwent an undirected coarsening, forming irregular shapes by merging with each other. When thermally exposed for over 10 h, the $\gamma'$ of the WQ and AC samples also gradually coarsened and lost its cubic shape. However, for the HT samples, it precipitated the TCP phase which surrounded by the $\gamma'$ envelope. Subsequently, the $\gamma'$ precipitates continued to coarsen and topological inversion was observed after 50 h of thermal exposure for all samples.

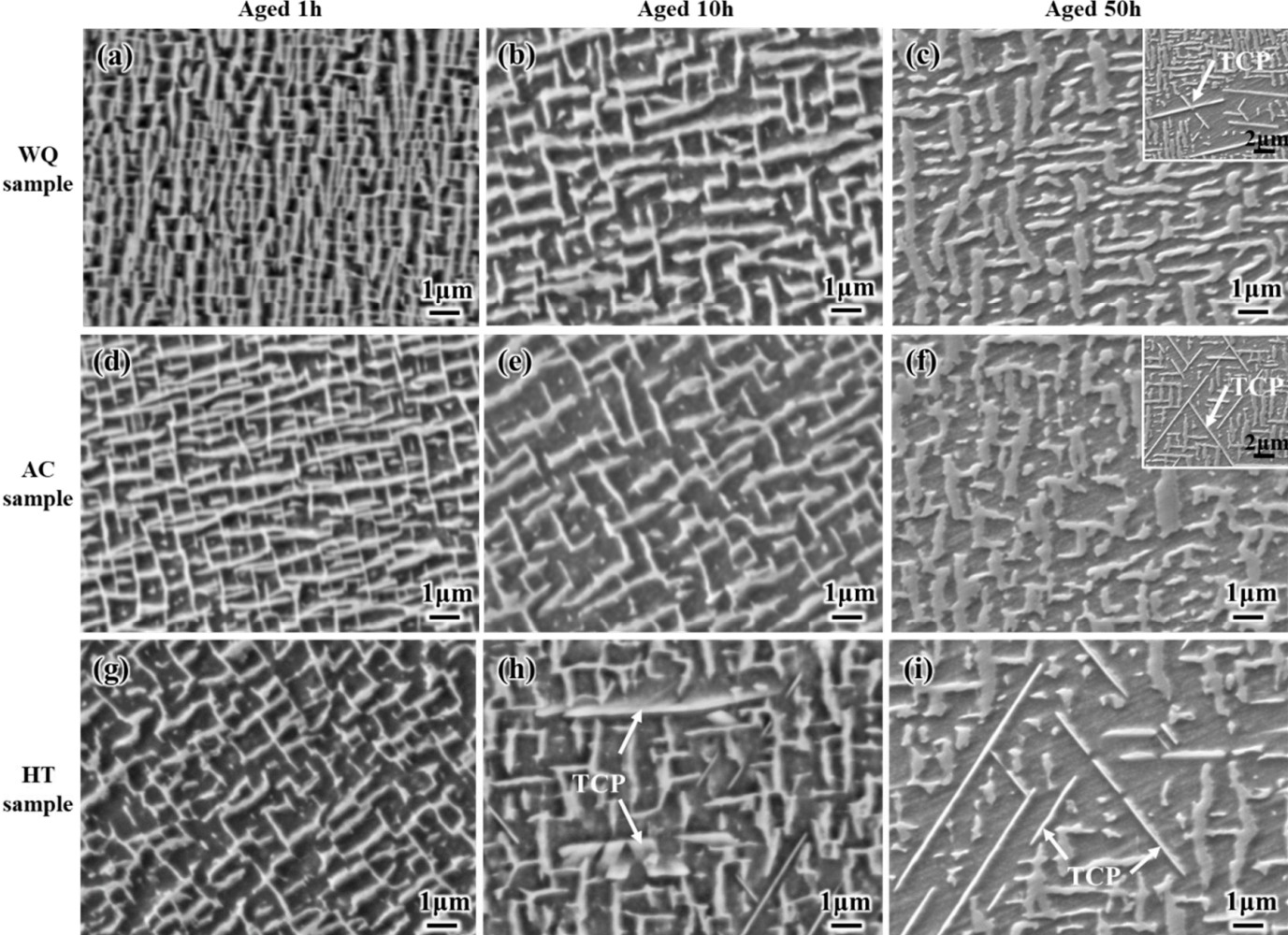

**Figure 7.** The SEM-SE microstructure evolution of $\gamma'$ phase after different heat treatment during thermal exposure at 1100 °C: (**a**–**c**) WQ samples; (**d**–**f**) AC samples and (**g**–**i**) HT samples.

The size of $\gamma'$ precipitates as a function of time is plotted in Figure 8a. It should be noted that the size of $\gamma'$ precipitates is not measurable after 50 h of thermal exposure due to the "labyrinth-like" shape. It can be seen that the size of the $\gamma'$ phase increased rapidly at first and then slowed down with the prolongation of the thermal exposure time. This was probably because the interfacial area and lattice mismatch strain of $\gamma/\gamma'$ phase, which is the thermodynamic driving force for the coarsening of the $\gamma'$ phase, was gradually consumed, so that the coarsening rate gradually decreased. Secondly, the WQ sample had the smallest $\gamma'$ phase size after thermal exposure for the same time, which can be explained by comparing the lower dominant coherent strain of $\gamma/\gamma'$ phase of the WQ sample to the other samples; in addition, during the coarsening process, more Re with low diffusion coefficients was repelled from the $\gamma'$ phase, which was beneficial to delaying the coarsening rate. Meanwhile, the coarsening of the $\gamma'$ phase broadened the spacing of the $\gamma$ matrix, and the $\gamma$ matrix width of the WQ sample increased the least, as shown in Figure 8b. In addition,

no significant tertiary γ′ phase was found in the γ-channels of all samples after 50 h of long-term thermal exposure, which means that the tertiary γ′ phase was dissolved back into the γ matrix as the γ-phase became coarsened and the elemental saturation decreased. After the long-term thermal exposure of 50 h, the area fraction of γ′ phase decreased and tended to be consistent for all samples, as shown in Figure 8c. However, at the initial stage of thermal exposure, both WQ and AC samples showed a transient increase in their γ′ phase area fraction rather than a single decrease for HT sample. This is because the initial stage of the thermal exposure process is similar to the aging process and allowed the γ′ phase to grow and reach the maximum area fraction.

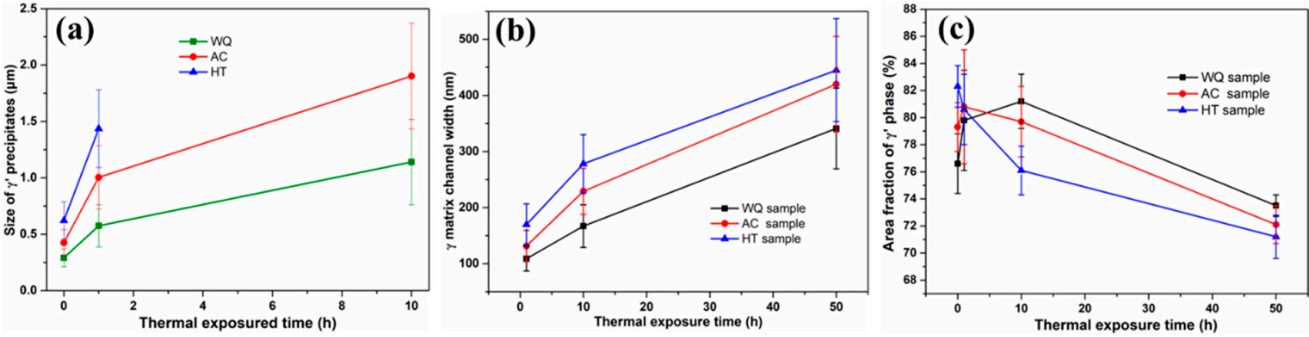

**Figure 8.** The Variation in the size of γ′ precipitates (**a**), width of γ matrix channel (**b**), and the area fraction of γ′ phase (**c**) in the alloys during thermal exposure at 1100 °C.

To illustrate the precipitation behavior of TCP during thermal exposure, firstly, the composition and structure of the TCP phase were analyzed, it was rich in Mo and Re elementsand identified as σ-phase [34], as presented in Figure 9. Then, the area fraction and length of TCP phases of the samples after different thermal exposure times were measured, as seen in Figure 10. It is clear that despite the as-cast samples undergoing the same solid solution heat process, the cooling rate after ST still significantly affected the precipitation behavior of the TCP phases. The area fraction of TCP phases in WQ sample after 50 h of thermal exposure was remarkably smaller than that in AC and HT samples, which was due to the fact that HT samples have the highest content of Mo and Re in their γ matrix, as listed in the previous Table 4. However, the length of the TCP phase in the WQ sample after 50 h of thermal exposure was higher than that in AC and HT samples, and the reasons for this will be discussed in detail in the next section. In summary, the faster cooling rate after ST not only slowed down the coarsening of the γ′ phase but also reduced the precipitation of the TCP phase, i.e., the fast-cooling rate was to some extent beneficial to the microstructure stabilization.

### 3.4. Creep Behavior under 1100 °C/137 MPa

The typical creep curves of the experimental alloy after different heat treatment are shown in Figure 11a. It can be seen that the creep process includes three stages, namely a primary stage, a steady state stage and an accelerated stage. The HT samples have the shortest rupture life of only 49.8 h. However, by increasing the cooling rate after ST, the rupture life of the WQ sample increased significantly from 101.2 h for the AC sample to 184.5 h, which was about four times longer than that of the HT sample. In addition, there was a slight increase in elongation of the HT sample compared to the WQ and AC samples. In order to better analyze the intrinsic mechanisms of the reduction in rupture life, quantitative features of creep behavior corresponding to different creep stages were researched, as illustrated in Table 5 and Figure 11b. The lowest steady state creep rate of $0.37 \times 10^{-4} \, \mathrm{h}^{-1}$ can be found for the WQ sample, which is about one-fifteenth of that of the HT sample; this is one of the reasons why it has the longest fracture life. It is noteworthy that the WQ sample has the highest strain and the longest time of the primary stage and tertiary stage, while the primary stage creep is almost invisible for the HT sample. Thus,

this suggested that increasing the cooling rate after ST not only affects the reprecipitation of γ′, but also influences the deformation mechanism of creep and significantly improves the creep properties of the alloy.

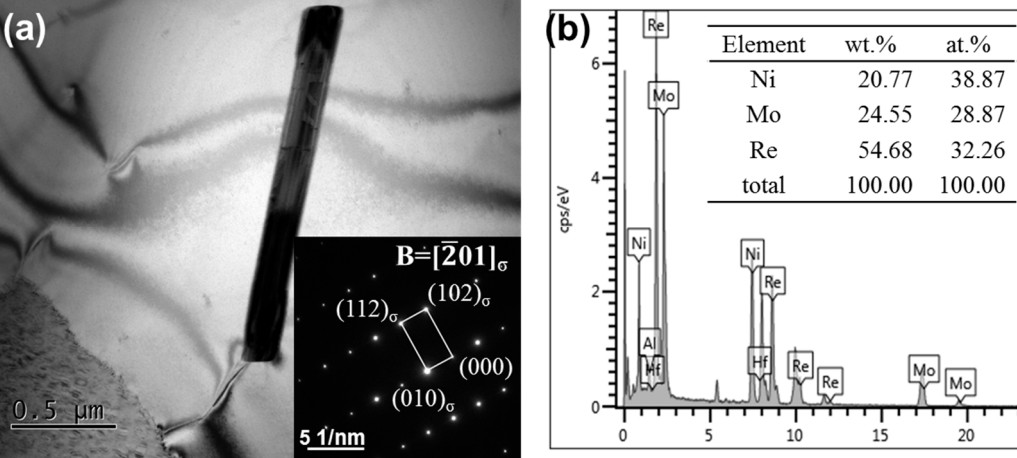

**Figure 9.** Typical TEM images and SAD patterns (**a**) and the compositional result by EDS (**b**) of σ phases for HT samples after thermal exposure at 1100 °C for 50 h.

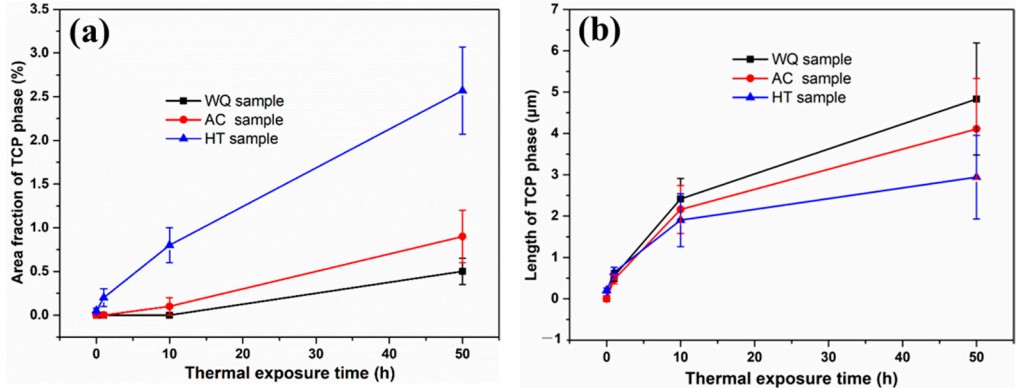

**Figure 10.** Area fraction (**a**) and length (**b**) of TCP phases of samples after different thermal exposure times.

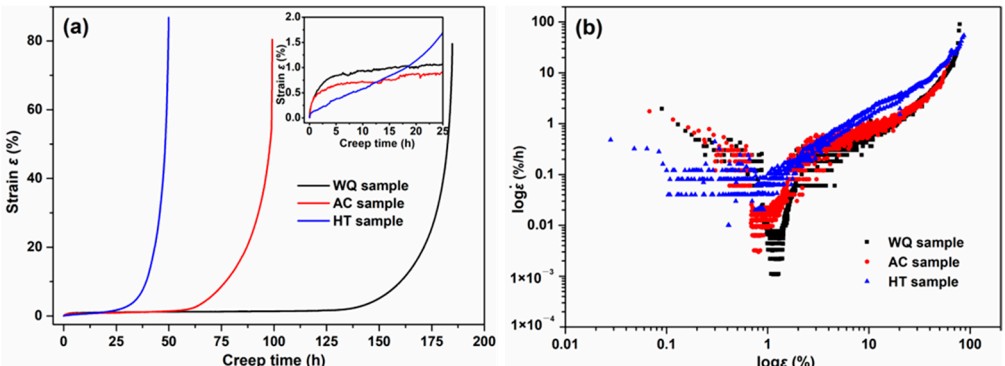

**Figure 11.** (**a**) Creep strain–time curves (The inset shows the primary stage of creep); (**b**) strain rate with respect to strain of the different heat treatment sample at 1100 °C/137 MPa.

**Table 5.** Creep properties of the different heat treatment sample at 1100 °C/137 MPa.

| Condition | Life (h) | Elongation (%) | Steady state Creep Rate $\dot{\varepsilon}_{ss}$ ($\times 10^{-8}$ $s^{-1}$) | Time of the Primary Stage (h) | Time of the Tertiary Stage (h) | Strain of the Primary Stage (%) |
|---|---|---|---|---|---|---|
| HT | 49.8 ± 4.6 | 80.0 ± 3.3 | 14.89 | 0.9 | 21.1 | 0.15 |
| AC | 101.2 ± 7.9 | 84.1 ± 4.5 | 4.02 | 8.9 | 36.7 | 0.71 |
| WQ | 184.5 ± 10.2 | 87.9 ± 5.2 | 1.03 | 15.4 | 57.4 | 0.98 |

The microstructures of samples after creep for 10 h and rupture at 1100 °C and 137 MPa are shown in Figure 12. There were significant differences in microstructure after 10 h of creep. The rafting was perpendicular (N-type rafts) to the loading direction and had been basically completed in the WQ sample, as shown in the Figure 12a. However, only part of the regions was rafted, and the morphology of the γ′ phase mostly was irregular in the AC sample. For the HT sample, its γ′ morphology is little different from that of the AC sample, but visible TCP phase precipitations can be seen. The WQ sample had a higher rafting rate and raft integrity compared to the other two samples due to its higher fraction of finer regular γ′ phase and the more negative lattice misfit obtained during the primary stage creep at a high temperature. In addition, more γ matrix, which seem like a "small island" and was embedded in the γ′ phase, were observed in AC and HT samplesed; this topological inversion microstructure restricted the motion of dislocations in the γ-phase and accounted for the lower strain in the primary stage creep.

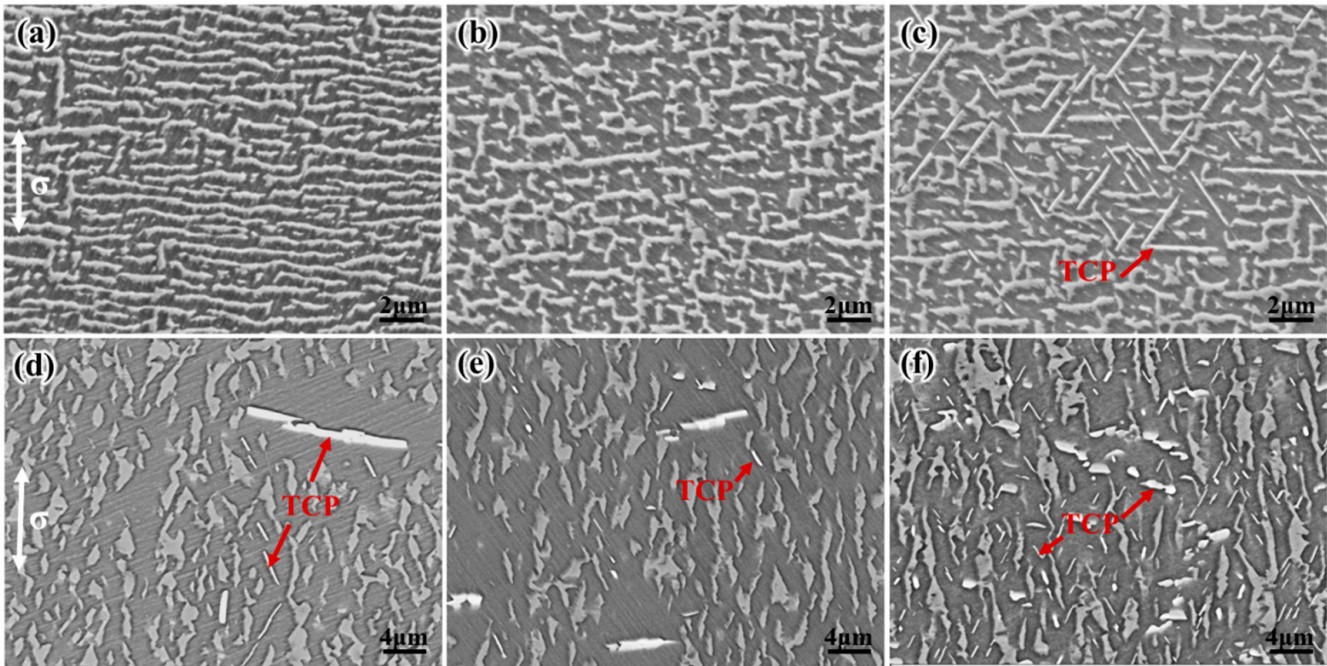

**Figure 12.** The microstructure of different samples after creep testing under 1100 °C/137 MPa: (**a**–**c**) interrupted for 10 h, (**d**–**f**) ruptured; (**a**,**d**) WQ sample, (**b**,**e**) AC sample and (**c**,**f**) HT sample (the loading direction is shown by the arrow).

The γ–γ′ interfacial dislocation network after crept interrupted for 10 h was observed, as shown in Figure 13. It can be seen that the dislocation network of the WQ sample is denser with the smallest spacing of about 22.3 nm, while the spacing of the AC sample and HT sample are about 33.4 nm and 40.6 nm, respectively. The order of magnitude relationship between them was consistent with the calculated results by Vegard Law, as shown in Figure 14. This is reasonable because for the WQ samples at the beginning of creep, the γ′ phase content increases under the thermal effect, and the elemental distribution between the two phases is aggravated. Particularly, the differences in Ta element content were

most pronounced (see in Table 6), which in turn makes it have a more negative mismatch degree and thus form a denser dislocation network. The dense dislocation network can effectively prevent the dislocations cutting into the $\gamma'$ phase, which is beneficial to improve the creep performance [35]. Therefore, increasing the cooling rate after ST is conducive to promoting the formation of rafts and forming a denser dislocation network during the primary stage of creep. After ruptured, the $\gamma$–$\gamma'$ phases microstructure of all samples undergoes more pronounced coarsening and loss of N-type rafts. The coarsened rattan shape $\gamma'$ phase is to some extent parallel to the stress axis due to the huge accumulation of deformation. In addition, the TCP phase presented in all samples, but the amount of TCP was significantly lower in the WQ samples compared to the HT, this coincides with the results of thermal exposure.

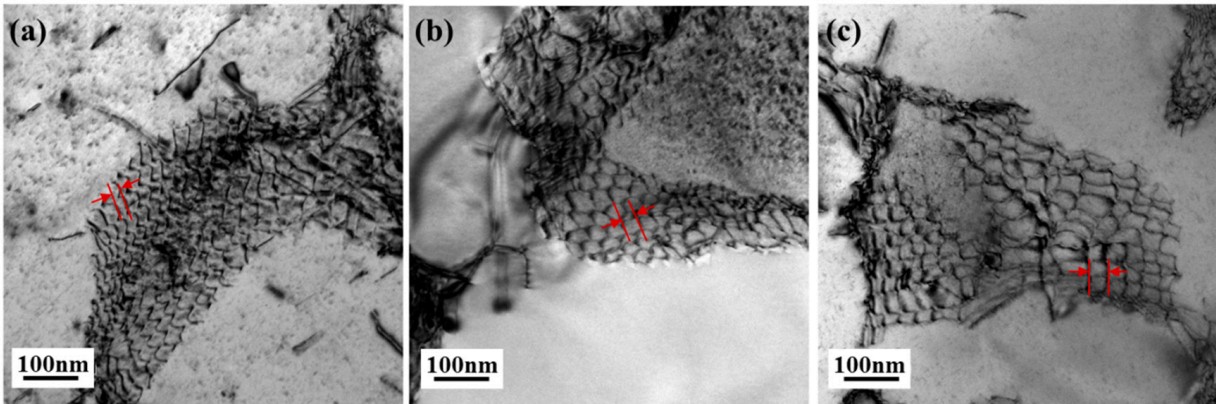

**Figure 13.** The dislocation network formed in samples interrupted by creep 1100 °C/137 MPa for 10 h: (**a**) WQ sample, (**b**) AC sample and (**c**) HT sample. (Beam = [001], the spacing of the dislocation network is marked by red arrows).

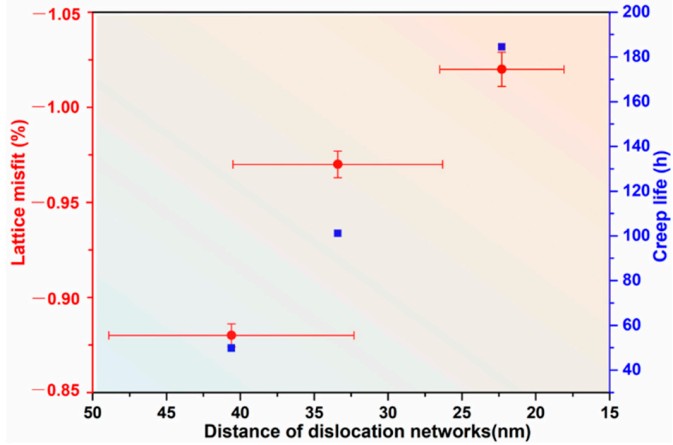

**Figure 14.** The lattice misfit and dislocation network spacing after 10 h of creep interruption at 1100 °C/137 MPa, and denser dislocation networks showed longer creep life.

**Table 6.** $\gamma$- and $\gamma'$ phase compositions after interruption of crept 10 h (at.%).

| Condition | | Al | Mo | Re | Ta | Ni |
|---|---|---|---|---|---|---|
| WQ | $\gamma$ | 3.57 | 13.84 | 4.03 | 2.36 | 76.19 |
| | $\gamma'$ | 12.62 | 4.55 | 0.36 | 2.70 | 79.78 |
| AC | $\gamma$ | 3.60 | 14.25 | 4.26 | 1.21 | 76.68 |
| | $\gamma'$ | 13.11 | 4.62 | 0.47 | 2.17 | 79.63 |
| HT | $\gamma$ | 3.68 | 14.06 | 4.35 | 1.13 | 76.79 |
| | $\gamma'$ | 13.02 | 4.70 | 0.50 | 2.25 | 79.55 |

Figure 15 shows the morphology in the longitudinal section of the AC and HT samples after creep deformation at 1100 °C/137 MPa. It can be seen that there were numerous micropores formed inside the fractured samples, and they decreased rapidly with the increase in the distance from the fracture surface, as shown in Figure 15a,b, and the quantitative statistical results are shown in Figure 16a. Micropores in SX superalloys generally include solidification pores (S pores), homogenization pores (H pores) and creep deformation pores (D pores). S pores and H pores are generally irregular or ellipsoidal (see Figure 15c), and the H pores are caused by the Kirkendall–Frenkel effect; meanwhile, the D pores are square- or rhombus-shaped, because its shape depends on the slip of dislocations on the octahedral slip system (see Figure 15d,e), and the long sides of the square is are parallel to the [001] direction, and its facets are lay on {110} planes. In this work, D pores accounted for the main part. Due to the formation of micropores, the true stress increased significantly, which in turn exacerbated the deformation of the nearby matrix to form more micropores. This is amongst the reasons why the creep rate increased rapidly at the end of creep.

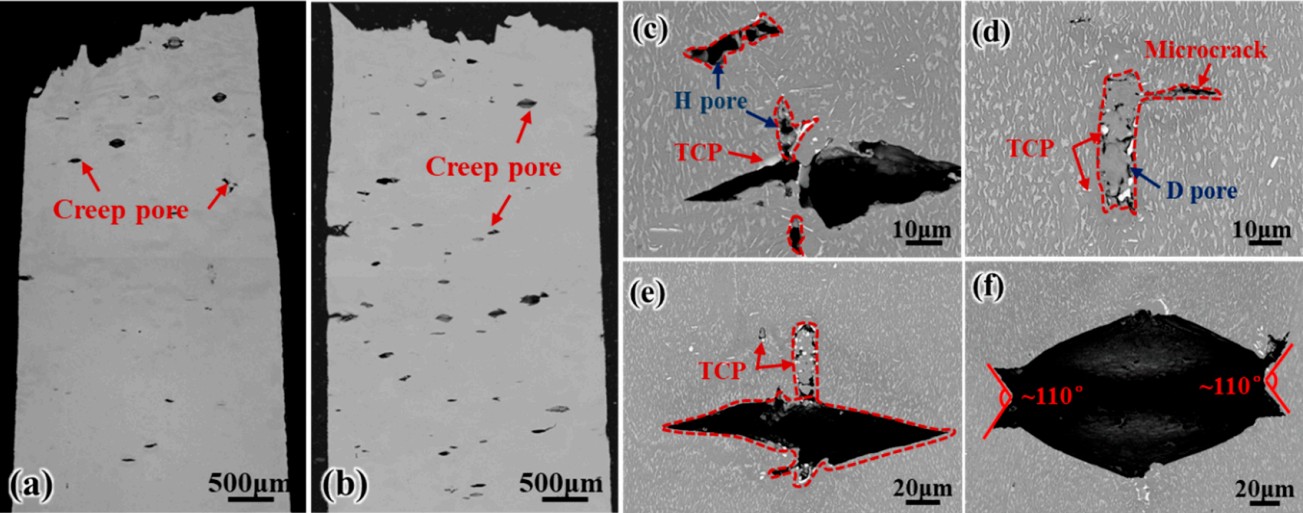

**Figure 15.** SEM micrographs of different samples ruptured after creep deformation at 1100 °C/137 MPa: (**a**) WQ sample, (**b**) HT sample, (**c**) TCP phases near homogenization pores, (**d**) TCP phases near the creep deformation pores and microcracks and (**e**,**f**) growth of pores and microcracks.

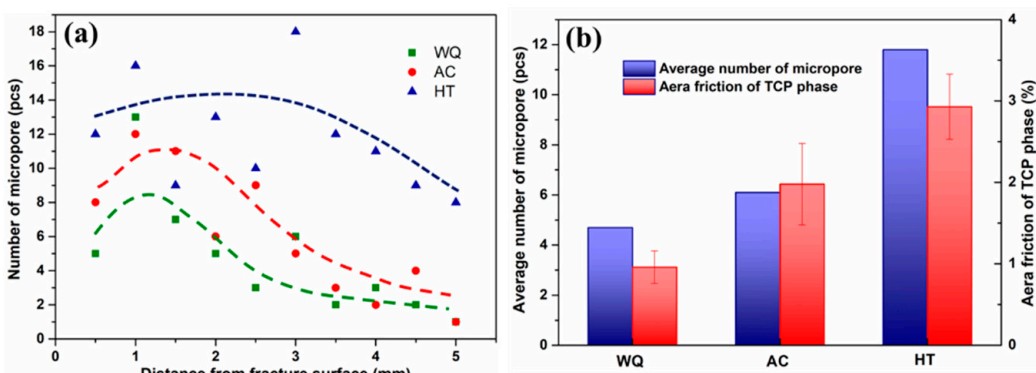

**Figure 16.** (**a**) number of micropores varies with the distance from the fracture surface and (**b**) histogram of average number of micropores and TCP phase area fraction.

Compared with HT samples, the number of micropores in WQ samples was significantly less, as shown in Figure 16a. Further observation of the microstructure around the micropores and microcracks revealed that a large number of TCP phases are accompanied

near them, as shown in Figure 15c–f. This implied that the micropores or microcracks were nucleated around the TCP phase and gradually grown with the accumulation of deformation. Actually, the area fraction of TCP phase of fractured WQ sample was about 0.9%, which was lower than 1.9% and 2.9% for the AC and HT samples, respectively. Thus, the average number of micropores was essentially proportional to the area fraction of the TCP phase after the ruptured samples is shown in Figure 16b. This explained why the WQ sample had the lowest number of micropores. In the early stage of creep, dislocations mainly moved in γ–γ′ phases microstructure, especially filled in the γ-channel. As the creep progressed, the γ–γ′ phases microstructure underwent directional coarsening, and the TCP phases gradually precipitated from the supersaturated γ-phase and grew up gradually. The TCP phase was brittle and semi-coherent with the γ matrix. If dislocations encounter the TCP phases when they are slipping or climbing in γ–γ′ phases microstructure, dislocations will accumulate at the interface of TCP/γ or TCP/γ′ and to generate vacancies. Because the easily movable dislocations are located in the <110> {111} of the octahedral slip system, it gradually grew into cuboid micropores along the [001] direction. Creep was continued, and the microcracks were easily initiated at the corners of the micropores due to stress concentration and propagated on the {111} planes which had the largest Schmid factor, as shown in Figure 15f. The microcracks and micropores grow continuously under the action of applied stress and connected with each other to form large cracks until fracture occurred. Thus, it was obvious that the initiation and propagation of these micropores and microcracks near the TCP phase accelerated the fracture of the SX superalloy.

## 4. Discussion

### 4.1. Effect of Cooling Rate on the Reprecipitation and Coarsening of γ′ Phase

In the previous section, the experimental results showed that the cooling rate after ST has a significant effect on the size distribution, morphology and phase composition of γ′ reprecipitation. In this research, upon solution treatment at γ′ super-solvus temperatures, the primary phase of alloy was completely dissolved and obtained a single γ solid solution, and the secondary γ′ phase precipitated during its subsequent cooling. A classical theory proposed by Ricks et al. [30] suggests that the morphological development of the γ′ reprecipitation from a single γ-phase solid solution occurs in the sequence spheres, cubes, arrays of cubes and eventually solid-state dendrites as coarsening is promoted by ageing, and this process is influenced by the lattice misfit [16,36]. For the experimental alloy, it has a high γ′ volume fraction (over 80 vol.%) and with a negative lattice misfit, so that it precipitates and transforms into a cubic secondary γ′ phase rapidly during cooling process after ST. When γ′ phases get coarsened, the coherent strain energy was increased and the γ′ phases were transformed to cubic with concave feature.

The difference in the nucleation and growth of γ′ reprecipitates caused by cooling rates cannot be ignored. The observed density value for the γ′ precipitates increased with the cooling rate, which indicated that the decomposition of the γ-phase occurred under large undercooling. Under large undercooling below the equilibrium instability temperature, the driving force for the nucleation of the γ′ phase is expected to increase rapidly, and thus the nucleation rate of the γ′ precipitate per unit volume will increase. The homogenous nucleation rate ($\dot{N}$) of the γ′ phase from the γ-phase was calculated from [37]:

$$\dot{N} = A exp\{\frac{-\Delta G^*}{kT}\} \exp\left\{\frac{-Q}{kT}\right\} \qquad (4)$$

where $A$ is a pre-exponential factor, $T$ is the temperature (K), $\Delta G^*$ is the activation energy for nucleation and $Q$ is the activation energy for self-diffusion. The activation energy for nucleation is calculated from:

$$\Delta G^* = \frac{16\pi\sigma^3}{3(\Delta G_v)^2} \qquad (5)$$

where $\sigma$ is the interfacial energy between $\gamma$- and $\gamma'$, $\Delta G_v$ is the driving force for $\gamma'$ precipitation. Babu et al. [24] calculated and summarized that the nucleation rate increases rapidly with large undercooling below the equilibrium instability temperature. However, below a certain temperature, the nucleation rate decreases slightly due to the reduction in atomic mobility. For the WQ condition, the density of the $\gamma'$ reprecipitates are 2 times higher than that of AC, as listed in Table 3. In addition, the shorter the diffusion time, the higher the particle number density, and the overlapping of precipitation diffusion fields limits the growth rate of $\gamma'$ precipitates under WQ conditions compared with AC conditions, resulting in the formation of finer size $\gamma'$ precipitates. Also, due to the limited diffusion rate, coupled with the superimposed fast cooling rate, there was only a very limited time for partitioning of the alloying elements between the $\gamma'$ and $\gamma$-phases. Therefore, the composition of $\gamma'$ precipitates was far from the equilibrium, as shown in Table 4. The high cooling rate during WQ prevented any further nucleation events despite the non-equilibrium conditions; thus, a monomodal size distribution of $\gamma'$ precipitates is observed. However, the relatively slower cooling for AC samples, the secondary $\gamma'$ precipitates nucleated at relatively higher temperatures (lower undercooling) with lower nucleation rates, resulting in a lower density of the precipitates, and the diffusion rate was fast enough to form a phase closer to the equilibrium composition. However, the $\gamma$ matrix region between the secondary $\gamma'$ precipitates, away from the growing $\gamma'/\gamma$ interfaces, retained a far-from equilibrium composition. As the temperature decreased during successive cooling, the driving force for $\gamma'$ phase nucleation increased, and eventually, re-nucleation at lower temperatures resulted in the formation of very fine tertiary $\gamma'$ precipitates in the region between the secondary $\gamma'$ precipitates.

The (directional) coarsening of the $\gamma'$ phase is a typical feature of the microstructure evolution of the superalloy under high-temperature thermal exposure, which have a significant effect on the high-temperature creep properties. Therefore, the coarsening rate of the $\gamma'$ phase in the alloy under different cooling conditions needs to be investigated. The size of $\gamma'$ and $\gamma$-phases continued to increase during thermal exposure at 1100 °C; the topological inversion phenomenon appeared even after 50 h due to the high $\gamma'$ phase fraction. The $\gamma'$ precipitates coarsening usually obeys the LSW theory based on matrix diffusion, the description for calculation is as follows [38,39]:

$$r_\text{t}^n - r_0^n = Kt \tag{6}$$

where $r_\text{t}$ is the average radius at the thermal exposure time $t$, $r_0$ is the average radius at $t = 0$, $n$ is the temporal exponent and $K$ is coarsening rate constant. Accordingly, the coarsening rate constants of $\gamma'$ precipitates in experimental alloys are presented in Figure 17. Obviously, the experimental data appeared to match perfectly to the classical LSW model when $n = 3$ according to Equation (6) for all samples. The $K$ values was 0.16 $\mu$m$^3$/h for WQ sample were calculated by fitting the slopes of the line connected by these points, which was much lower than the values for the AC and HT samples. So, the $\gamma'$ coarsening rate was significantly reduced by increasing the cooling rate after ST.

The coarsening of the $\gamma'$ phase is an important reason for the decline of the creep performance. In order to strengthen the high-temperature creep performance of the superalloy, it is necessary to slow down the coarsening process of the $\gamma'$ phase as much as possible. In general, the coarsening of $\gamma'$ precipitates depends on the reduction in the interfacial energy and coherent strain energy of the $\gamma$–$\gamma'$ phases. Under the condition that the volume fraction of the $\gamma'$ phase is basically unchanged, the interfacial energy of the $\gamma$–$\gamma'$ phases decreases with the increase in the size of the $\gamma'$ precipitate. Furthermore, the coherent strain energy is related to the distribution of alloying elements in the $\gamma$- and $\gamma'$ phases; the lower the microsegregation of the $\gamma/\gamma'$ phase is, the smaller the coherent strain energy is. Because the interfacial energy is difficult to tune by considering the formation of fine and cubic $\gamma'$ precipitates to sufficiently hinder the movement of dislocations, the coherent strain energy can be achieved by reducing the microsegregation of the $\gamma/\gamma'$ phase. From the results in Figures 7, 8 and 17, it can be determined that by increasing the cooling rate after solution

treatment, the microsegregation of the two phases was decreased (see Table 4), and the coherent strain energy was also lowered. Although the interface energy of $\gamma/\gamma'$ phase increased somewhat due to the reduction in the size of the $\gamma'$ phase, the overall driving force for coarsening of the $\gamma'$ precipitate was decreased. In addition, in the coarsening process, more rhenium with a low diffusion coefficient needed to be repelled from the $\gamma'$ phase, which together slowed down the coarsening rate of its $\gamma'$ phase.

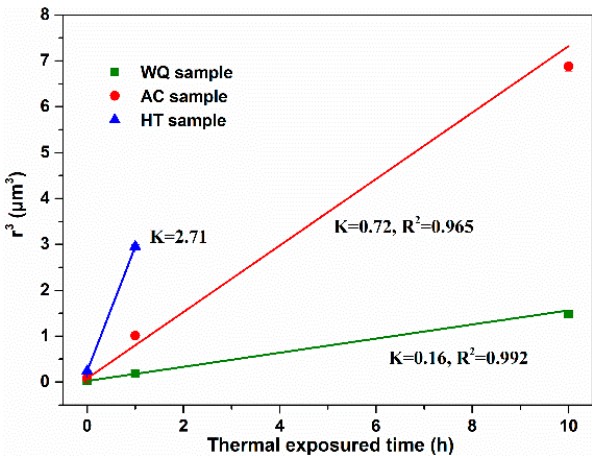

**Figure 17.** Plots of precipitates size ($r^3$) vs. time ($t$) for $\gamma'$ phases during thermal exposure at 1100 °C, the coefficient of determination ($R^2$) were also labeled.

### 4.2. Effect of Cooling Rate on Precipitation of the TCP Phase

As has been established, the TCP phase precipitates from the supersaturated $\gamma$-phase during thermal exposure [13,40] and causes severe degradation of the alloy properties [9,41]. On the one hand, the precipitation of the TCP phase consumes a large amount of solid solution elements, such as Mo and Re, leading to a decrease in the strength of the $\gamma$ matrix; on the other hand, TCP as a brittle phase causes stress concentration and becomes the origin of micro-cracks [34]. Therefore, the supersaturation of alloying elements in $\gamma$-phase of heat-treated specimen determined the nucleation and growth behavior of the σ-phase. As mentioned above, the Mo and Re concentrations of the $\gamma$-phase in the WQ samples under rapid cooling are lower than those in the AC samples under slow cooling, as shown in Table 4. So, a lower nucleation rate of the σ-phase is observed in the WQ specimens than in the AC specimens during the initial period of thermal exposure, which is consistent with the lowest TCP phase area fraction of the WQ sample in Figure 10a. However, for the HT samples, which under a slow cooling rate and having undergone two-step aging, the $\gamma'$ phase adequately grew and expelled Mo and Re elements from the $\gamma'$ phase into the $\gamma$-phase. This further increased the supersaturation of alloying elements in the $\gamma$-phase and thus rapidly precipitated a large amount of σ-phase during the initial period of thermal exposure. Nevertheless, with the growth of the σ phase, the Mo and Re concentrations of the $\gamma$-phase in HT samples decreased significantly after the initial period of thermal exposure due to the remarkable consumption of Mo and Re elements by σ phase, while the growth rate of the TCP phase is proportional to the supersaturation degree of the $\gamma$-phase during thermal exposure [40]. Therefore, there were lower Mo and Re concentrations of $\gamma$-phase in the HT samples compared to the WQ samples, which resulted in a reduction in growth driving force of σ-phase and thus shorter σ phase. Altogether, the cooling rate after ST changes the supersaturation degree of the $\gamma$-phase and thus affects the nucleation and growth behavior of the TCP phase. This phenomenon was probably attributed to the fact as follows: first, with the temperature decreased from upon the $\gamma'$ dissolution temperature, the $\gamma'$ phase precipitated from the supersaturated $\gamma$-phase and it's a diffusion-type phase transformation, so the composition of the $\gamma$- and $\gamma'$ phases is influenced by the cooling rate. Increasing the cooling rate after ST, the time and temperature for diffusion of alloying elements are reduced and the elements are 'frozen' in $\gamma$- and $\gamma'$ phases. Therefore,

the compositional difference between the γ- and γ′ phases in the WQ sample is smaller, i.e., the Mo and Re content (saturation) of the γ-phase is lower.

*4.3. Effect of γ′ Reprecipitation and TCP Phase on Creep Properties*

It has been mentioned above that the creep properties of the experimental alloy were substantially improved by increasing the cooling rate after solid solution. It is essential to investigate the relationship between the microstructural evolution (γ–γ′ phases and TCP phase) and the stress-rupture property of the alloy. In general, the main strengthening mechanisms of Ni-based single crystal superalloy include solid solution strengthening [42,43], precipitation phase strengthening [44] and interfacial dislocation network strengthening [45]. At high temperatures, the mechanical properties of SX superalloys are strongly dependent on the morphology, size and volume fraction of the γ′ phase. A classical work by Nathal [46] showed that in a Re-free Ni-based single crystal superalloys with a negative mismatch, the creep life shows a peak at a γ′ phase size of about 450~500 nm, and the optimal size range of γ′ phase decreases as the lattice misfit becomes more negative. Murakumo [5] suggested that the highest creep life was obtained with a γ′ volume fraction of about 70% at room temperature. It can also be demonstrated that the alloys containing spherical precipitates had lower creep strain rates than those containing cubic precipitates [7]. Generally, the strength increments in critically resolved shear stress (CRSSs) for the precipitates shearing ($\Delta\tau_c$) is calculated using followings:

$$\Delta\tau_c = \frac{1}{2}\left(\frac{Gb}{\lambda}\right)\frac{2w}{\pi}\left(\frac{2\pi\gamma_{APB}r}{wGb^2} - 1\right)^{0.5} \tag{7}$$

and the stress $\sigma_{OR}$ required for a dislocation to slip through a narrow matrix channel is given by equation:

$$\sigma_{OR} = \sqrt{\frac{2}{3}}\left(\frac{Gb}{hS}\right) \tag{8}$$

where $\gamma_{APB}$ is the antiphase boundary energy of γ′ phase, $b$ is the Burgers vector, $r$ is the γ′ mean size, $\lambda$ is the average interparticle spacing, $G$ is the shear modulus, $h$ is the width of the γ-channel, $w$ is the fitting parameter and $S$ is the Schmid factor. In the current study, the experimental alloys used water cooling after solution treatment, which had the smallest γ′ phase size and γ-channel width. In the early stage of creep, $\sigma_{OR}$ is the dominant due to the dislocations move mainly in the γ-channel, thereby the WQ sample has the strongest creep strength at this stage. After that, the γ and γ′ phases are coarsened to a certain extent (about 550~700 nm), and its γ′ volume fraction increased (about 80%), as seen in Figure 8. At this time, the strength increment $\Delta\tau_c$ brought by the γ′ phase is dominant. Therefore, the WQ sample always has the highest strength compared to the other samples.

Furthermore, it can be seen that the faster cooling rate samples have a larger strain of primary creep stage. Rae [47] proposed that the primary creep rate is controlled principally by the dislocation activity and density accumulating of <112>{111} slip systems in the γ-channels, and the greater primary creep is due entirely to a more rapid creep rate. During the primary creep stage, for the WQ sample, a regular and cubic γ′ phase was enveloped by γ-channels due to the faster cooling rate after ST, whose γ-channels are interconnected and dislocation-depleted state; the dislocation of <112>{11} slip systems is easily activated and accumulated, and coupled with a longer primary creep stage can accumulate a higher strain. However, for AC and HT samples, its γ′ phase is coarser and irregular, and there are tertiary γ′ phases or pre-formed dislocation networks in the γ-channel. The presence of fine tertiary γ′ particles impede propagation of the perfect matrix dislocations which are forced to cut the γ′ precipitates much earlier in some localized regions [48]. The strain of primary creep is particularly influenced by tertiary γ′ precipitation in the γ-channel because of these particles may increase the resistance of the normal dislocation bowing out and dislocation motion is limited [49], thus ultimately leading to a low strain. However, it will undergo dissolution during elevated temperature thermal exposure and therefore

affects more the primary creep behavior rather than the whole creep process. This is also in agreement with the fact that no significant tertiary $\gamma'$ was found in the microstructure of thermally exposed and creep-interrupted samples.

Under elevated temperature and low stress, the original cubic-shaped $\gamma'$ phase undergoes directional coarsening and the higher the degree of lattice misfit between $\gamma$- and $\gamma'$ phases, a denser dislocation network is formed at the interface of the two phases, which prevented the dislocations cutting into the $\gamma'$ rafts, thus improving the creep lifetime [10]. The microstructure turns into a more complete rafts perpendicular to the stress for the WQ sample compared to the AC and HT sample. In addition, a denser dislocation network formed in the WQ sample can effectively prevent dislocations from penetrating into the $\gamma'$ phase; the stability of dislocation network helps to correspondingly stabilize the lamellar $\gamma/\gamma'$ structure during the stable creep stage. Thus, the WQ sample has the lowest steady state creep rate. Furthermore, when the creep enters the tertiary stage of creep, the grown TCP consumes a large amount of Re and Mo and reduces the solid solution strength of the matrix, and the TCP phase becomes the nucleation site for micropores and microcracks because it is a brittle phase and semi-coherent with the matrix, which accelerates the damage accumulation at tertiary stage. Compared with the HT and AC samples, the rapidly cooled sample (WQ) had a smaller amount of TCP and delayed its precipitation, which not only delayed the onset of entry into the tertiary stage, but increased the time of the tertiary stage. These are representative of the reasons as to why faster cooling rates lead to longer creep life of the superalloy.

## 5. Conclusions

1.  The cooling rate after solid solution treatment significantly affected the size distribution and morphology of the $\gamma'$ reprecipitates (secondary $\gamma'$ phase), and with the increasing of the cooling rate, the size of $\gamma'$ precipitates decreases while its number density increases. The morphology of $\gamma'$ phase also transforms from cubic to complex shape.
2.  The size of $\gamma'$ reprecipitates under fast cooling rate showed a monomodal size distribution, while the tertiary $\gamma'$ phase precipitated in the $\gamma$-channel under slow-cooling rate, and the faster cooling rate leads to lower Mo and Re content in the $\gamma$-phase (i.e., lower supersaturation degree).
3.  During the thermal exposure at 1100 °C, the coarsening rate of $\gamma'$ phase under fast cooling rate is slower, and the area fraction of the precipitated TCP phase is lower due to lower supersaturation degree of the $\gamma$-phase, but the length of the TCP is longer compared to the slow-cooled sample.
4.  The creep life of the alloy significantly improved by increasing the cooling rate after solution treatment. On the one hand, this comes from the better homogeneity and thermal stability of the microstructure under high cooling rate after solution heat treatment; On the other hand, The finer cubic $\gamma'$ precipitation facilitated rafting, which, combined with the reduction in the TCP phase as the origin of microcracks, resulted in better high-temperature creep properties of the alloy.

**Author Contributions:** Methodology, J.H.; validation, Y.P.; formal analysis, H.Z. and C.A.; investigation, J.H.; writing—original draft preparation, J.H.; writing—review and editing, C.A., Y.R. and J.H.; supervision, S.L. and S.G.; project administration, S.G.; funding acquisition, Y.S., S.G. and H.Z. All authors have read and agreed to the published version of the manuscript.

**Funding:** This research was funded by National Science and Technology Major Project of China [Grant numbers: 2017-VI-0016-0088 (S. Gong), 2019-VI-0016 (Y. Shang)]; the National Natural Science Foundation of China (NSFC) [Grant number: 52101117 (H. Zhang)].

**Institutional Review Board Statement:** Not applicable.

**Informed Consent Statement:** Not applicable.

**Data Availability Statement:** Not applicable.

**Conflicts of Interest:** The authors declare no conflict of interest.

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
