# Peer review of "The Effect of Cooling Rate from Solution Treatment on γ′ Reprecipitates and Creep Behaviors of a Ni-Based Superalloy Single-Crystal Casting"

_crystals, doi:10.3390/cryst12091235_

Round 1

Reviewer 1 Report

Dear Authors

Your paper provides some interesting results, useful for researchers working on single-crystal castings, and it is worth to be published. However some corrections are required so please take into consideration following remarks:

Title – there is mistake in the title – it should be “from” instead of “form”, and I strongly suggest to write “solution treatment” (then it makes sense). Moreover, some Ni-based superalloys are called the “single-crystal” (e.g. CMSX-4), but actually the casting can be single-crystal one. I suggest to write “… a Ni-based superalloy single-crystal castings”

Line 24 – “Ni-based single crystal (SC) …” – The abbreviation “SC” seems obvious, but it is widely accepted to use the “SX”. Moreover, the term “single-crystal casting” should be add to the “Keywords”.

Lines 28 – “Ni3Al” – please be careful with subscripts.

Lines 92 and 93 – “The nominal composition of the experimental superalloy (Exp. Alloy) was listed in Table 1.”? The nominal composition rather refers to commercial alloys and it is given by the producer. In case of an experimental alloy I suggest to use the term “chemical composition” and to provide the method of its determination.

Lines 96-98 – “The single-crystal rods were achieved by high-rate solidification (HRS) …” sounds too general. Process parameters are crucial for production of SX castings. What was the size of the rods? Have you checked the porosity? (especially you analyse it after the heat treatment) How did you check that it was definitely SX rods having with this orientation (“… within 5° deviating from the [001] orientation.”).

Lines 103-106 and Table 2 – “SHT” – according to what standards is this recommended heat treatment for examined alloy (which is an experimental one)? What kind of protective atmosphere did you use?

Multi-stage ST and AT are really recommended by manufacturer of aviation parts – do you think your variants (AC and WQ) could be accepted by aerospace industry?

Line 134 – “as-casting”? – however, the term "as-cast" is used more often.

Line 180 – Fig. 2a is so small that it doesn’t show much.

Reviewer 2 Report

This manuscript studied the cooling rate on the properties of Ni-based superalloy. Some question are listed as below.

1.     What is the meaning of low-density (Line 89), I do not think it is a low-density Ni-based alloy.

2.     What are the dimensions of the single-crystal rods? It should be described in the manuscript. Also, the authors should provide the clear evidence to prove that they have a single crystal structure.

3.     Fig.2 is too small to observe, and nothing can be found from these images.

4.     The scale bars in Fig.7 are too small.

5.     Fig.15 only shows the cross-section images of the specimens after creep test. It is better to provide the SEM images of fracture surfaces of the specimens after creep test into this manuscript.

6.     Some editing errors, such as Ni3Al (Line 28), L12 structure (Line 29), backscattered electron image (BEI, Line 153). The authors should prepare the manuscript carefully.

Reviewer 3 Report

The proposed manuscript is interesting; there are many weaknesses that need to be improved. This based on the following:

·        It is recommended not to use acronyms in the Abstract without first defining them

·        The abstract should be reviewed again because it is very general and the objective is not clear.

·        The scope of the study is not well defined, the authors could better express it in the abstract

·        The formation of the gamma phase does not depend only on the cooling rate, but also the alloying elements as well as the austenitization temperature and the residence time.

·        In the keywords section, Nickel is symbolized with Ni

·        The introduction is very poor, it should be reinforced better.

·        In the introduction section the objective is very extensive, it should be more specific

·        Table 1: The authors must indicate by which technique the nominal composition was obtained

·        The microstructure of figure 1 must be integrated into the results section

·        The authors must specify if the microstructure of figure 1 was obtained by optical or scanning electron microscopy and what is the magnification used.

·        Table 2. What is the meaning of the acronyms AC, WQ and SHT, it is necessary to indicate it

·        Line 127, in SEM which sensor uses secondary or backscattered electrons?

·        In section 2. Materials and Methods, subsections should be included to help the reader to better understand the structure of the article.

·        Figures 2 and 3 should start for example Figure 1. SEM-BSE morphology .......

·        Figure 5. The microstructures obtained by TEM should be better explained.

·        Some of the paragraphs in the results section are very long and should be better distributed

·        The results of table 4. Were obtained by what techniques, EDS?

·        Figure 6. X-ray spectra are very fuzzy and unclear.

·        Fig. 7. The microstructures of figure 7 were obtained by OM, SEM or TEM

·        I believe that the authors should explain better how is the variation of the size of the precipitates of the gamma phase.

·        the values obtained by creep reported in table 5 must integrate the standard deviation

·        Are the values in table 6 the result of the EDS analysis?

·        The authors could better explain the mechanism of microcracks and pores in creep tests.

·        The discussion of results should be enriched, only the authors describe the results but they are not discussed.

·        In the conclusions section the first paragraph must be deleted it is not a conclusion

·        More current references from 2019,2020,2021 and 2022 must be included

·         The authors present 54 references, there is no self-plagiarism

Round 2

Reviewer 2 Report

I agree with this revised version of manuscript.

Reviewer 3 Report

The authors have made the necessary changes, for which the manuscript can be accepted for publication.